# The Impacts of Chinese FDI and China–Africa Trade on Economic Growth of African Countries: The Role of Institutional Quality

**Miao Miao [1]** , **Qiaoqi Lang [1], Dinkneh Gebre Borojo [1],\*, Jiang Yushi [1,2] and Xiaoyun Zhang [1]**

[1]  School of Economics and Management, Southwest Jiaotong University, Chengdu 610031, China; miaomiao@swjtu.edu.cn (M.M.); joccilqq@gmail.com (Q.L.); jys_a@sina.com (J.Y.); annezhang@swjtu.edu.cn (X.Z.)
[2]  Service Science and Innovation Key Laboratory of Sichuan Province, Chengdu 610031, China
\*  Correspondence: g10dinkneh@yahoo.com

**Abstract:** While there is a consensus on the expanding importance of the China–Africa economic relationship, there is much more debate on how to portray the relationship. Thus, this study is aimed to examine the impacts of the China–Africa trade and Chinese foreign direct investment (FDI) on the growth of African countries controlling the mediating role of institutional quality. The two-step system Generalized method of moments (GMM) model is applied using robust data for the period of 2003–2017. Drawing on complementary theoretical perspectives, this study took into account the conditional effect of China–Africa trade and Chinese FDI subject to the institutional quality of African countries and the interdependence of China–Africa trade and Chinese FDI to African countries. The benign impacts of the China–Africa trade and Chinese FDI on economic growth to African countries remain contingent upon appropriate policy action to improve the institutional quality of African countries and the synergies between the China–Africa trade and Chinese FDI to African countries.

**Keywords:** foreign direct investment (FDI); trade; growth; institutional quality; two-step system GMM estimator; China; Africa

## 1. Introduction

The impacts of international trade on economic growth remain the prominent issues in both theoretical and policy context. The nexus of trade and economic growth has gained even more attention in recent years, considering the persistent and widespread differences in economic performance among countries, especially among developing countries in the wake of growing international trade integration (Silajdzic and Mehic 2018). However, the empirical analyses are as inconclusive as the theoretical perspectives. For example, the study by Chang et al. (2009) and Zahonogo (2016) indicate that the effects of foreign trade on economic growth are conditional to the different structural characteristics of the economy.

However, according to a study by Kim (2011), foreign trade has significant adverse effects on developing countries. Furthermore, Baliamoune-Lutz (2011) and Matthew and Folasade (2014) found that international trade only had little significance on economic growth and implied that the domestic institutions' weakness is responsible for enhancing the growth effect of international trade.

Likewise, the impact of foreign direct investment (FDI) on economic growth is inconclusive. The FDI spillover to the host economy is significantly determined by the economy's ability to benefit

from FDI. Absorptive capacity indicates an economy's capacity to absorb the opportunities spilled over by the FDI inflow (Khordagui and Saleh 2013). Hence, the role of FDI to enhance economic growth will depend on the domestic absorptive capacity of FDI-receiving countries (Khordagui and Saleh 2013; Jyun-Yi and Chih-Chiang 2008; Jude and Levieuge 2016; Ali et al. 2010).

The increasing China–Africa economic relations received much attention and debate among scholars. It is revealed by the China–Africa trade, and the FDI flow from China to African countries. The literature on the China–Africa relationship often falls under two broad categories. Some argue that the relationship is a win–win for African countries and China. However, some others view the relationship as a new form of imperialism, especially concerning the issue of African countries' resource exploitation (Adisu et al. 2010; Foster et al. 2008; Kolstad and Wiig 2011).

More particularly, a study by He (2013), using manufactured goods exports of sub-Saharan Africa to represent the production of countries, reveals that Africa's trade relation with China has a positive and significant effect on the economic development of African countries. However, the benefit in terms of trade effect is limited to those African economies that export natural resources (Busse et al. 2016). On the other hand, for other scholars, the trade between Africa and China has an insignificant or a negative effect on the economic growth of African countries and is more dominated by China's economic interests, namely the access to critical resources (Foster et al. 2008; Adisu et al. 2010; Busse et al. 2016). Moreover, as it is clearly maintained, some researchers argue that Africa's trade with China is centered on China's interest in Africa and deteriorating effect on the African economy.

Similarly, the impact of Chinese FDI on the economic growth of African countries is controversial. According to Foster et al. (2008), Adisu et al. (2010) and Busse et al. (2016), FDI flows from China to African countries seem to play no major role in African countries' economic growth and are perceived to follow the state-driven strategy of giving infrastructure and taking natural resources. On the other hand, the Chinese foreign direct investment flow indicates China's significant role in African countries' economic performance. For example, the study by Pigato and Tang (2015) shows those African countries which have benefited from the China–Africa economic relationship, though the bulk of Chinese investment is focused on a few resource-rich countries. Furthermore, Whalley and Weisbrod (2012) and Doku et al. (2017) indicated that an increase in China's FDI stock and flow into African countries has a positive effect on African countries' economic growth.

These studies, however, are not sufficient, not without limitations, and no study was conducted using appropriate econometric specifications and considering the domestic capabilities of African countries to reap growth effect of the China–Africa economic relations. As a result, we can find different empirical literature that reveals that the effects of the China–Africa trade and Chinese FDI on the economic growth of African countries are mixed at best. Furthermore, it is interesting to note that no attempts have been made to investigate the impact of Chinese FDI and China–Africa trade on African countries' economic growth from the perspective of institutional quality and their role in mediating the relationship. Thus, this study's unique contribution towards the body of knowledge is that it has extended the classical economic model to capture the moderating role of institutional quality in the nexus of the China–Africa economic relationship (China–Africa trade and Chinese FDI) and economic growth of African countries.

Therefore, the major aim of this study is to analyze the impacts of the China–Africa trade and Chinese FDI on the economic growth of the African countries, controlling the mediating role of the institutional quality of African countries.

The rest part of this paper is organized as follows. Part two discusses the stylized facts of Chinese FDI and the China–Africa trade, the institutional environment and economic performance of African countries. Part three presents the literature review and hypothesis. Part four explains the data and methodology of the study. Part five presents the results and findings of the study. Part six presents a detailed discussion of the study, and part seven gives conclusions based on the findings.

## 2. Chinese FDI and China–Africa Trade, Institutional Quality and Economic Growth of African Countries

In the last two decades, Africa registered a period of sustainable and impressive economic growth. Some of the countries can be included among the fastest-growing economies in the world. Most African countries have registered a sustainable economic growth with an economic growth rate exceeding 5% per year over the past 15 years. The average growth rate of the sample African countries is presented in Appendix F. The growth registered in this period was supported by favorable external conditions, especially high commodity prices boom as a result of high international demand for commodities, and the increase in investment money and FDI flow in search of new economic opportunities at the global level have played a significant role (Zamfir 2016). Among external factors, a vital change that took place in Africa was the rise of Brazil, Russia, India, China and South Africa (BRICS countries) and their strengthening economic and trade relationships with African countries. While one should be careful in establishing a relationship between the emergence of the BRICS economies in Africa and Africa's recent impressive economic growth record, the BRICS have certainly contributed to changing perceptions of Africa's economic fortunes among the international donor community. The most important emerging player from the BRICS group on African soil is China because China provides the largest share of foreign finance for Africa from the South–South cooperation (Woods 2008).

China has a long trade relationship with African countries. However, the fastest growth of the trade between African countries and China was registered in the last two decades. According to trade data from the Global Trade Atlas, trade between China and Africa has increased at the fastest rate since 2000. The total trade between Africa and China has registered a compound growth rate of 24.7% in the last two decades. It also implies that the trade in goods and services has grown at 26.7% annually on average for the past two decades (Borojo and Yushi 2016).

Furthermore, Chinese FDI is distributed in almost all African countries. This reveals that Chinese outward FDI to African countries plays a prominent role in China's economic interaction with many African countries (Miao et al. 2020). The distribution of Chinese FDI in sample African countries is reported in the Appendix D.

Additionally, the institutional environment of African countries is weaker and needs improvement. The mean score of the sample African countries' institutional quality indicators indicates that the African countries continue to score below expectations with the negative scores outweighing the positive ones. For instance, the reports in Appendix E suggest that, on average, the political environment (−0.478) across the sample is quite unstable. This indicates that there is still political instability and violence, which taints the democratic atmosphere of the countries. The worst performer on the scale, which has proven resistant to change, is government effectiveness, which is still lagging behind with an average score of negative 0.651. Thus, the governments' ability to offer quality public services and to formulate and implement policies that are indicative of government effectiveness is low. This is closely followed by a rule of law with the average score of negative 0.621. Furthermore, the regulatory quality and control of corruption, which are under the due attention of international organizations, has recorded the average negative score of 0.565 and 0.562, respectively (see Appendix E). Therefore, the scores of the institutional quality of African countries indicate that the countries' performance in the institutional environment is weak and needs policy attention.

## 3. Literature Review and Hypothesis Development

The impacts of the China–Africa trade and Chinese FDI remain to be one of the significant issues in empirical studies. This section focuses on the theoretical and empirical foundations of the trade between China and African countries and Chinese FDI, and provides a hypothesis based on economic theories and empirical studies. Furthermore, it gives a detailed discussion on the mediating role of institutional quality in the relationship between economic growth and FDI and trade.

### 3.1. China–Africa Trade and Economic Growth

The relationship between international trade and economic growth remains one of the most important issues in theoretical and empirical studies. The nexus of international trade and economic growth has given due attention in the recent period in view of the persistent and widespread differences in economic performance among different countries, especially among developing countries in the wake of growing international trade integration (Silajdzic and Mehic 2018).

The necessary foundation for the theoretical linkage between international trade and economic growth are Smith's absolute advantage and Ricardo's comparative advantage principles of the classical school of thought, as well as the Heckscher–Ohlin (H–O) resource endowment trade model that bases trade among nations on the differences of resource abundance.

Based on Smith's absolute advantage principle, benefit from international trade and specialization is maximized when one country has an absolute cost advantage (that is, uses less labor to produce a unit of a product) in one good and the other nation has an absolute cost–benefit in the other good. On the other hand, according to Ricardo's comparative advantage principle, even if a country takes no absolute cost-benefit in the production of both commodities, a source for mutually valuable trade may still happen. The relatively less efficient country should specialize in and export that commodity in which it is relatively less inefficient (where its absolute disadvantage is the least) and the relatively more efficient country should specialize in and export that good in which it is relatively more efficient (where its absolute advantage is the greatest) (Krugman et al. 2012).

Furthermore, based on the H–O resource endowment (factor abundance) trade model, a nation will have a comparative advantage in specializing and exporting that commodity for which a large amount of relatively abundant (cheap) input is used; and will import that commodity in the production of which the relatively scarce (expensive) input is used (Krugman and Obstfeld 1997). Hence, according to the H-O model, the benefit from trade for countries is fundamentally based on the differences in the abundance of factors of production (inputs).

Additionally, the endogenous growth theory has led to an extensive inventory of new hypotheses that stress the significance of international trade in achieving a sustainable rate of economic growth. This theory has emphasized the degree of openness, terms of trade and export performance, to confirm the assumption that open economies grow more rapidly than closed ones (Edwards 1998). Likewise, the advocates of the export-led strategy and free trade point out that most developing countries that followed inward-oriented policies under the import-substitution strategy (ISS) had poor economic achievements (Balassa 1978). Thus, many countries were forced to stimulate their export-led orientation because most of them had to rely on multilateral organizations to implement adjustments to correct imbalances in their fundamental macroeconomic indicators. The strategy was to encourage a free market through policies that relied heavily on the export promotion approach as one of the most suitable and trustworthy mechanisms. Consequently, by the mid-1980s, the economic literature concerning development economics, economic growth, and the adjustment strategy had quickly rejected the inward-oriented approach and was suddenly placing great emphasis on the export-led strategy.

The impacts of trade, specifically export, on economic growth, are based on the export-led growth notion affect through enhanced economies of scale, advanced technology adoption, and higher capacity utilization. In particular, export growth increases investment in those sectors in which a country has a comparative advantage, increasing the national output and raising the rate of economic growth (Feder 1983; Al-Yousif 1997). Furthermore, in their theoretical models, Grossman and Helpman (1991) show that trade among countries can enhance the transfer of new technologies and facilitate countries' productivity improvement. That is, in addition to its direct effect on capital accumulation, trade among countries can positively impact the economic performance of trading countries through channels such as technology transfers, scale economies and comparative advantage. Therefore, it can generate economic growth by facilitating the diffusion of knowledge and technology from the direct import of high-tech goods.

In line with the aforementioned theoretical literature, a number of empirical studies have been conducted on the relationship between foreign trade and economic growth. Nonetheless, the empirical literature is as inconclusive as to the theoretical framework in the nexus of trade and growth. A study by Chang et al. (2009) researched the effect of trade on economic growth for some developed and developing countries. Their findings indicate that the effects of trade on economic growth are subject to the different macroeconomic and structural characteristics of the trading economies. The results further reveal that the effect of trade openness on economic growth for sub-Saharan Africa might be conditional on other complementary variables such as educational investment and infrastructure.

In addition, using a dynamic growth model with data for 42 countries covering from 1980 to 2012, Zahonogo (2016) assesses how trade openness affects economic growth in sub-Saharan African countries. Furthermore, Matthew and Folasade (2014) conducted a study on the effect of trade openness, institutions and economic growth in sub-Saharan Africa and found that trade openness only had a little significant influence on economic growth in the sub-Saharan African countries.

The impact of the China–Africa trade on the economic growth of African countries is controversial. Some argue that China's growing trade with Africa is useful to African countries' economic growth (Chemingui and Bchir 2010; Baliamoune-Lutz 2011). In contrast, others claim that the China–Africa trade which continues to be promoted by China's rising profile does not correspond to the region's longer-term aims, that is, to diversify its economic and trade framework and ensure that trade contributes to the overall and industrial development of African countries (Ademola et al. 2009). Thus, the impact of the China–Africa trade on economic growth for African countries can be positive or negative. However, the institutional quality of countries can help absorb the positive effect of trade openness. Thus, improving institutional quality is an essential aspect of developing economies. The establishment of strong institutions is as crucial as a good macroeconomic policy environment. Promoting economic openness to the rest of world in the presence of poor institutional quality as characterized by corruption, weak rule of law, ineffective government and regulatory quality, and poor contract enforcement policies may encourage more rent-seeking behavior by the bureaucrats, resulting in an adverse effect on the economic performance of the countries. In other words, weakness in the institutional environment diverts resources from productive sectors to the less productive and rent-seeking sectors (Kandiero and Wadhawan 2003).

**Hypothesis 1 (H1).** *The impact of China–Africa trade on the economic growth of African countries is positive, conditionally to the institutional environment of African countries.*

*3.2. Chinese FDI and Economic Growth*

The previous segment contained the association between economic growth and foreign trade. In this segment, the nexus of FDI and growth is highlighted. Using the Heckscher–Ohlin (H–O) trade model to explain the motives behind investors who operate production chains abroad in the 1960s and internalization theory, Dunning developed the OLI paradigm. The OLI paradigm consists of three sub-paradigms from which one can analyze the reasons why firms engage in FDI: ownership (O), location (L), and internalization (I). These determinants have categorized into three types: market-seeking, resource-seeking, and efficiency-seeking (Dunning 2000). Furthermore, Helpman et al. (2004) developed a theory that relates FDI to international trade. On the other hand, Nocke and Yeaple (2008) developed an assignment theory to analyze the volume and composition of foreign direct investment.

It is a commonly accepted claim that FDI is an essential instrument in the rapidly changing international economic integration and relationship among countries. It is also linked to globalization. FDI believed in providing a base for creating direct, stable, and long-lasting links between different economies in the world. Under the sound policy framework, FDI can serve as an essential instrument for local enterprise development. It may also help promote the competitive position of both the host countries (recipient) and the investing (home) economy. It also provides a chance for the host economy

to enhance its products more broadly in international markets. FDI, in addition to its positive impacts on the development of international trade, is an essential source of capital for a range of host and home economies (OECD 2008).

The theoretical explanations of FDI largely stem from international trade that is based on the theory of comparative advantage and differences in factor endowments between countries. FDI is usually attracted to a particular country by the comparative advantage that the country or region offers. FDI refers to the long-term participation by a country in another country, and this involves participation in management, joint-venture, transfer of technology, and expertise.

The impacts of FDI flow on economic growth have been given due attention in different economic kinds of empirical literature. It is argued that FDI has essential effects on the economic development of receiving countries. From a theoretical point of view, FDI can impact the domestic economies of host countries via capital accumulation (Titarenko 2005). The nexus of FDI-growth has been explained in different theoretical frameworks. FDI is particularly essential for developing countries since it provides access to resources that would otherwise be unavailable to these countries.

According to the neo-classical exogenous growth model of Solow (1956, 1957), economic growth is a function of the labor force and capital, keeping technology exogenous. This model states that capital accumulation contributes to the economic growth of countries. In this framework, therefore, FDI can directly impact economic growth through capital accumulation and the inclusion of new inputs in the production function of the host country (Mahembe and Odhiambo 2014).

Hence, based on the above theoretical literature, a number of empirical studies were conducted to analyze the impacts of FDI on countries' economic growth. For example, Dollar and Kraay (2004) examined the interrelation between international trade, FDI, and poverty. They used an estimation model with GMM and instrumental variable methods for more than 100 countries. Their period was ten-year averages over the 1970–2000 period. They found that FDI affects the well-being of people positively by increasing their income and decreasing poverty.

Some scholars argue that the impact of FDI can be positive if it targets the growth of host countries, and their institutional quality is better and does not depend on resource extraction. For example, Nunnenkamp and Spatz (2003) argue that FDI might have positive impacts on growth if a host country does not receive resource-seeking FDI. According to their results, the host country and industry nature, as well as the interplay between both sets of characteristics, have a significant implication on the growth impact of FDI in developing countries.

According to some literature such as Asheghian (2004) and Hansen and Rand (2006), the positive impact of FDI on the host countries economic growth depends on certain factors that exist or not in those countries, such as human capital, the degree of openness of its economy, the economic and technological conditions, political stability and the absence of violence.

Therefore, the FDI spillover to the host economy is significantly determined by the economy's ability to benefit from the FDI. Absorptive capacity indicates an economy's capacity to absorb the opportunities spilled over by FDI inflow. Absorptive capacity factors mediate FDI spillovers that influence an economy's ability to absorb the effect of FDI (Khordagui and Saleh 2013). Hence, the role of FDI to enhance economic growth will depend on the domestic absorptive capacity of FDI-receiving countries.

For example, Jyun-Yi and Chih-Chiang (2008) conducted a study on the impacts of FDI on economic growth, using threshold regression techniques developed so that they could examine whether the effect of FDI on economic growth was determined by the different domestic absorptive capacities of FDI-receiving countries. They used initial GDP, human capital and the volume of trade as threshold variables in their study. Some empirical studies show that FDI alone plays an ambiguous role in contributing to economic growth, depending on the threshold variables.

Furthermore, Khordagui and Saleh (2013) conducted a study on FDI and domestic absorptive capacity in emerging economies, including some African countries. They examined the effect of FDI, conditionally on human capital, trade openness, and the institutional quality of the FDI-receiving

countries. The findings of the study refer that FDI spillovers exist in emerging Middle Eastern and North African countries, yet they are more evident when controlling for schooling as an absorptive capacity factor. The findings reveal that both the trade openness and domestic institutional quality of the sample countries appear to have an insignificant influence on the FDI spillovers, for emerging and emerged Middle Eastern and North African economies. Moreover, their results reveal that countries with lower schooling levels stand to benefit the most from FDI spillovers. Jude and Levieuge (2016) investigate the effect of FDI on economic growth, conditionally on the institutional quality of host countries. They developed several theoretical arguments to show that institutional heterogeneity may be an explanation for the mixed results of previous empirical studies. Using a panel smooth regression model on a large sample of developing countries, they revealed that FDI flow has a significant positive effect on growth, only beyond a certain threshold of institutional quality. Hence, to benefit from FDI-led growth, institutional reforms should thus precede FDI attraction policies. Additionally, they believe that some reforms seem to advocate faster marginal effects of FDI, while institutional complementarities may result in an incremental effect on growth. Likewise, using panel data for a number of countries from 1981 to 2005, a study by Ali et al. (2010) analyzes the significance of institutional quality in determining foreign direct investment (FDI). Their results indicate that institutional quality indicators are robust determinants of FDI flow and that the most noteworthy institutional quality features are connected to legal protection and propriety rights. However, the institutional quality of African countries is weak compared to the rest of the regions across the globe. The performance of African countries regarding the institutional quality environment is weak, scoring the negative scores. On average, the political environment across the African countries is unstable and subject to violence. This indicates that political instability and violence should still have policy attention and needs to be improved. Furthermore, African countries have the worst performance in government effectiveness, implying that the governments' ability to provide quality public services and to formulate and implement policies that are indicative of government effectiveness is relative. Likewise, the rule of law is also weak and needs to be retained to enhance economic relationships with other countries, specifically China. Additionally, the regulatory quality and the government's ability to control corruption are weak (See Appendix E).

Therefore, based on the theoretical and empirical literature, the following hypothesis is articulated.

**Hypothesis 2A (H2A).** *The impact of Chinese FDI on the economic growth of African countries is conditional to African countries' institutional environment.*

**Hypothesis 2B (H2B).** *The China–Africa trade mediates the impact of Chinese FDI on economic growth.*

## 4. Methodology and Data

This section discusses the strategy used to examine the impacts of the China–Africa trade and Chinese FDI on the economic growth of African countries and the data used for the exercise.

### 4.1. Method of Analysis

This part mainly focuses on the discussion of the strategy to analyze the effects of the China–Africa trade and Chinese FDI on African countries' economic growth. To this end, a dynamic panel data model was employed to analyze the impacts of the China–Africa trade, Chinese FDI flow on the African economy, extending the neoclassical augmented growth model developed by Mankiw et al. (1992) that relates growth with different explanatory variables. The following part briefly describes the functional specification of the empirical models that are used to analyze the impact of trade between Africa and China and Chinese FDI flow from China to Africa on the economic growth of African countries.

The analysis of the effect of the China–Africa trade and Chinese FDI flow on African countries' economic growth was based on a pooled data set of cross-country and time-series observations, using a sample of 44 African countries for the period of 2003–2017. Hence, this is one of the reasons to use

the two-step system GMM estimator of Arellano and Bover (1995), employed to examine the impacts of the China–Africa trade and FDI flow on the economic growth of African countries.

There are several reasons to use a dynamic panel data model (the two-step system GMM model) to exercise the growth equation. First, the two-step system GMM system estimator is employed for the estimation of the effect of the trade openness of China–Africa and the inflow of Chinese FDI and aid to Africa on the economic growth of African countries because our analysis is based on a pooled data set of cross-country and time-series observations.

Second, the economic growth by itself is a dynamic phenomenon that can be captured by dynamic panel models. Hence, to determine this dynamic phenomenon, the system GMM model is suitable. Third, the two-step system GMM model can address the endogeneity problem and gives consistent estimates, even in the presence of measurement error. Fourth, this model provides asymptotically efficient inference, assuming a minimal set of statistical assumptions (Arellano and Bover 1995; Blundell and Bond 1998).

In general, the system GMM is an estimator designed for conditions with small time (T) and large panels (N). It is a systematic method with a few periods and many individual units with a linear functional relationship, one left-hand variable that is dynamic, depending on its past realizations, and right-hand side variables that are not strictly exogenous; associated with past and possibly current realizations of the error fixed individual effects, implying unobserved heterogeneity (heteroskedasticity) and autocorrelation within individual units' errors, but not across them.

In the two-step system GMM model, the China–Africa trade, Chinese FDI and financial aid variables are considered as endogenous because there will be a reverse causality problem between economic growth and China–Africa trade, and economic growth and financial aid and economic growth and FDI inflow from China to African countries.

Therefore, using this dynamic model helps us by providing an important robustness check on the statistical inferences and yield accurate estimates. It is sufficient to exploit the full use of the information contained within the data set. Based on the above information, the econometric equation that relates the dependent variable (real gross domestic product (real GDP) per capita growth for African countries) to China–Africa trade, Chinese FDI inflow to Africa and control variables is specified in the following equation:

$$g_{it} = \beta_0 + \beta_n X_{it} + \delta_m I_{it} + \phi_n T_{it} + \delta_n IN_{it} + \varepsilon_{it} \tag{1}$$

where $g$-denotes the per capita real GDP growth of African countries, $X_{it}$ represents the control variables such as the lag of per capita GDP growth of the countries, physical capital, the inflation rate of African countries, health expenditure of African countries and the institutional quality of African countries. $I_{it}$ is a Chinese FDI in African countries. $T_{it}$ is trade between China and African countries. $IN_{it}$ represents the interaction terms among institutional quality and the China–Africa trade, institutional quality and Chinese FDI, and China–Africa trade and Chinese FDI. $\varepsilon_{it}$ indicates the stochastic term.

The logarithm-linear transformation of Equation (1) is:

$$g_{it} = \beta_0 + \beta_n \ln X_{it} + \delta_m \ln I_{it} + \phi_n \ln T_{it} + \delta_n \ln IN_{it} + \varepsilon_{it} \tag{2}$$

In this model, one of the innovations of this study is the inclusion of the conditional effect of the China–Africa economic relationship on the economic growth of African countries in which the previous empirical studies lack. The conditional effect of the China–Africa trade and Chinese FDI to the domestic absorptive capacity of African countries on economic growth was incorporated in the model. Hence, the interaction terms of the China–Africa trade, Chinese FDI as the Chinese foreign trade, Chinese FDI, foreign aid, service contracts and labor cooperation were closely mixed and combined. Hence, the impact channels of Chinese trade, FDI and aid overlap (McCormick 2008).

To see the interaction, for example, consider the Chinese FDI in the mining and construction sectors. These types of FDI were often accompanied by loans from the Chinese government to African governments, which in turn contract Chinese firms to extract resources or build infrastructure. In turn,

the repayment of these loans is often tied to commodity exports from African borrowers to China. As a result, the interaction term is included in the model specification.

*4.2. The Data*

This study used data for the period 2003–2017 based on the data availability for 44 African countries. The sample countries included in this study are reported in Appendix C. Trade data between African countries and China were taken from the tralac database. The GDP of African countries, gross fixed capital formation, inflation, the domestic credit to private sector per GDP, mobile subscription rate and health expenditure were compiled from World Development Indicators (WDI). Chinese FDI to Africa data were compiled from United Nations Com Trade (UNCTAD) bilateral FDI Statistics and the China Statistical Yearbook by the Johns Hopkins China–Africa Research Institute. Financial aid data from China to African countries were obtained from the AidData website.

Institutional quality indicators such as the control of corruption, absence of violence and security, rule of law, government effectiveness, voice and accountability and regulatory quality of African countries were collected from Worldwide Governance Indicators (WGI) World Bank Group website. They feature the scores for various dimensions of institutional quality by combining information from several independent sources. They extract information from all these sources, transform their scales so that every indicator is constructed as a normally distributed random variable with a zero mean, unit standard deviation, and ranging approximately from −2.5 to 2.5 with higher values showing better institutions. Using these six indicators of the institutional quality of African countries, we derived a single composite indicator using principal component analysis. From Table A2, the eigenvalues of the first principal component of governance and institutional quality is greater than 1 (4.725 > 1). However, none of the other components have eigenvalues more than 1. Since the first component explains 78.8% of the original variables' variation, the study uses the eigenvectors of the first principal component as weights in constructing an institutional and governance index (see Appendix B).

## 5. Results and Findings

This section mainly focuses on the impacts of the China–Africa trade and Chinese FDI to African countries on the economic growth of African countries. It provides the results and findings of the study. The summary statistics of the major variables are reported in Appendix A. Table 1 presents the two-step system GMM estimates of the effects of the China–Africa trade and Chinese FDI on the growth of African countries.

The regression command (xtabond2) of a Roodman (2009) dynamic panel data model is applied because it can do everything xtabond does and has many extra characteristics. The second lag is used in the specification because the second lag is not correlated with the current error term, but the first lag is correlated with the current error term. In addition, it is suggested to use second or deeper lags to get a good instrument, but using deeper lags can decrease the sample size. The conventional covariance matrix is robust in a two-step system GMM estimation of panel-specific auto-correlation and heteroskedasticity, but the standard errors are biased downwards (Roodman 2009). STATA 13 is used to exercise the regression.

The institutional quality index is derived using the governance quality index derived from six governance indicators (control of corruption, absence of violence and instability, government effectiveness, regulatory quality, rule of law, voice and accountability) using principal component analysis. Additionally, the diagnostic test of the model was conducted to evaluate the robustness of the model. Thus, the Hansen test is conducted for the over-identification issue. We fail to reject insignificant statistics for the Hansen test for the two-step system GMM in Table 1. Therefore, it confirms that the instruments satisfy the orthogonality condition, or all instruments are valid in the model. Similarly, the Wald test for the joint significance of the variables is conducted and does not reject our model specification. The serial correlation test of the Arellano and Bond test is applied and does not reject the null that there is no second-order serial correlation. These diagnostic tests, therefore, validate the use

of the two-step system GMM model to analyze the impacts of the China–Africa trade and Chinese FDI flow on the growth of African countries.

**Table 1.** The impacts of the China–Africa trade and Chinese foreign direct investment (FDI) on the economic growth of African countries.

| Variables | I | II | III | IV |
|---|---|---|---|---|
| lnloglaggrowth$_{it}$ | −0.172 *** (0.022) | −0.192 *** (0.022) | −0.228 *** (0.017) | −0.240 *** (0.018) |
| lncapital$_{it}$ | 0.186 *** (0.010) | 0.168 *** (0.012) | 0.129 *** (0.009) | 0.156 *** (0.016) |
| lninflation$_{it}$ | −0.215 *** (0.028) | −0.248 *** (0.019) | −0.282 *** (0.029) | −0.276 *** (0.025) |
| lnhealth$_{it}$ | −0.112 *** (0.018) | −0.104 *** (0.025) | −0.151 *** (0.015) | −0.109 *** (0.019) |
| lnChinaFDI$_{it}$ | −0.307 ** (0.141) | −0.268 ** (0.110) | 0.111 (0.143) | −0.165 (0.110) |
| lntradeChina_Africa$_{it}$ | 0.045 (0.058) | 0.043 (0.031) | −0.083 (0.066) | −0.001 (0.041) |
| lninstitution$_{it}$ | 0.094 *** (0.026) | 0.096 *** (0.031) | 0.179 *** (0.015) | 0.178 *** (0.029) |
| lninstitution*FDI$_{it}$ | | 0.008 ** (0.004) | | |
| lninstitution*trade$_{it}$ | | | 0.020 *** (0.003) | |
| lntrade*FDI$_{it}$ | | | | 0.009 *** (0.003) |
| _cons | −4.257 *** (0.193) | −3.805 *** (0.263) | −2.830 *** (0.169) | −3.557 *** (0.377) |
| Obs. | 477 | 477 | 468 | 477 |
| Groups | 40 | 40 | 40 | 40 |
| Wald (p-value) | 0.000 | 0.000 | 0.000 | 0.000 |
| AB2 (Ch2-sta) p-value | 0.46 | 0.466 | 0.408 | 0.382 |
| Hansen (Ch2sta) p-value | 0.839 | 0.864 | 0.807 | 0.896 |

Notes: *** significant at 1%, ** significant at 5%, standard error in parenthesis, AB2test is Arellano and Bond tests for autocorrelation, Hansen test is the test for over-identification, Wald: overall fitness of the model test, Obs.: observations, Group: group of countries. Per capita real GDP growth of African countries is the dependent variable, laggrowth$_{it}$ represents the lag of growth, ChinaFDI$_{it}$ represents Chinese foreign direct investment in African countries, tradeChina_Africa$_{it}$ denotes the trade between China and African countries, Capital$_{it}$ denotes the physical capital proxied by gross fixed capital formation, inflation$_{it}$ shows the inflation rate of African countries to proxy the macroeconomic stability of African countries, health$_{it}$ indicates the human capital of African countries proxied by gross secondary school enrolment, institutions$_{it}$ is the institutional quality of African countries that proxy domestic the absorptive capacity of countries.

Among the traditional determinants of growth, physical capital and institutional quality are significant determinants of growth in African countries. Their coefficients are positive and statistically significant. The accumulation of physical capital, therefore, enables economic growth for African countries. In the development process, thus, physical capital accumulation can be a primary engine for economic growth. On the other hand, the coefficients of inflation and health expenditure are negative and statistically significant. Maintaining price stability by controlling inflation is a robust factor for economic growth. However, the coefficient of credit to the private sector has a negative and significant effect on the GDP per capita growth of African countries. Furthermore, the coefficient of the lag of per capita GDP is negative and statistically significant.

Among the variables of interest, the results in Table 1 indicate that trade between China and African countries has an insignificant effect on the GDP per capita growth of African countries when controlled individually in Column (I) (Table 1).

On the other hand, the coefficient of Chinese FDI is negative and statistically significant, indicating a negative association between Chinese FDI and GDP per capita growth when we consider it individually. This might be because the growth effect of FDI should be backed by the domestic adaptive capacity

to use the benefit of Chinese FDI to excel in economic growth. That is, Chinese FDI flow to African countries alone is not sufficient for African countries to reap the benefits of FDI. The positive impacts of FDI on host countries' economic growth depends on whether certain factors exist or not in those countries, such as the degree of openness of its economy, political stability, the absence of violence or domestic institutional quality (Asheghian 2004; Hansen and Rand 2006). Furthermore, in a recent empirical investigation, McCloud and Kumbhakar (2011) noted the existence of a heterogeneous relationship between FDI and economic growth across developing countries. They argued that, across countries, differences in institutional quality were correlated with heterogeneous absorptive capacities and hence, a heterogeneous FDI–growth relationship. Hence, the role of FDI to enhance economic growth will depend on the domestic absorptive capacity of FDI-receiving countries. It is based on this that we further examined the conditional effect of FDI on domestic institutional quality.

Therefore, based on the justification, we extended our analysis to the conditional effects of FDI and China–Africa trade with institutional quality because different variables can complement the growth effect of FDI flow. As a result, we used the interaction terms of FDI to the institutional quality of African countries to determine its conditional impact as the improvement in institutional quality complements FDI flow and the China–Africa trade in enhancing economic growth.

Furthermore, Chinese FDI flow can also be affected by trade between China and African countries. We include the interaction terms of China–Africa trade and FDI as the Chinese foreign trade and FDI have been closely mixed and combined. The impact channels of Chinese trade, FDI and aid overlap affect each other (McCormick 2008). Hence, we control the interaction terms of China–Africa trade and Chinese FDI in our specification. Because of the high correlation among the interaction terms, we separately control the interaction terms from Column (II) to Column (IV).

Accordingly, the conditional effect of Chinese FDI flow to African countries and China–Africa trade to domestic institutional quality is strongly positive. Therefore, the results are in line with the hypotheses (H1 and H2A) (see Table 2). This result implies that institutional reforms can promote the marginal effects of Chinese FDI and China–Africa trade as institutional complementarities may lead to an incremental effect on growth. To benefit from FDI-led growth and the China–Africa trade, therefore, institutional reforms should precede FDI attraction and trade policies. This result is supported by some literature such as Khordagui and Saleh (2013) and Jude and Levieuge (2016) that reveal FDI spillovers exist in different economies, yet that they are more evident when controlling for the domestic absorptive capacities of countries.

**Table 2.** Hypothesis test.

| Parameter | Estimate | Std. Err. | z-Value | *p*-Value | Decision |
|---|---|---|---|---|---|
| H1 (lninstitution*trade$_{it}$) | 0.020 | 0.003 | 5.94 | 0.000 | supported |
| H2A (lninstitution*FDI$_{it}$) | 0.008 | 0.004 | 2.14 | 0.033 | supported |
| H2B (lntrade*FDI$_{it}$) | 0.009 | 0.003 | 2.86 | 0.004 | supported |

Notes: ins.trade$_{it}$, ins.fdi$_{it}$ and ins.aid$_{it}$ are the interaction terms of the China–Africa trade and institutional quality, Chinese FDI and the institutional quality and Chinese aid and institutional quality, respectively.

Furthermore, the China–Africa trade has a growth-enhancing effect on Chinese FDI and the growth nexus of African countries. This result indicates that trade openness can play a role in facilitating the spillover of FDI benefits to the host country. The findings are in line with the FDI literature that argues that FDI and trade openness can be complementary for economic growth and that an economy promoting more open trade policies, especially export promotion policies, are more likely to benefit from FDI spillovers (Nobakht and Madani 2014). Therefore, the China–Africa trade openness can be a mediating factor in the impacts of Chinese FDI flow on African countries' economic growth. Thus, the result supported the acceptance of our hypotheses (H2B).

Furthermore, we run for other African countries excluding South Africa, Nigeria, Zambia and Sudan to partly reduce reverse causality issues and to look at if the China–Africa trade and Chinese FDI

have the same relationship with the whole sample. South Africa is one of the largest hubs for Chinese FDI. The volume of trade, FDI and financial aid allocation from China to these countries is larger than the rest of African countries. South Africa is the largest trading partner for China in Africa, while China has been the country's biggest trading partner accounting for a quarter to a third of the China–Africa overall trade. It is the top country exporting to China and importing from China. According to FDI Markets, South Africa is the first destination for Chinese direct investment. Similarly, Nigeria, Sudan, Zambia and Algeria are the top destinations of Chinese FDI next to South Africa. About 60% of Chinese FDI stock in Africa was allocated to these countries (see Appendix D). Furthermore, these countries have a big share of trade with China. Therefore, the exclusion of these countries helps analyze the sensitivity of the results. The results are reported in Table 3 below.

**Table 3.** The impacts of the China–Africa trade and Chinese FDI on the economic growth of African countries.

| Variables | I | II | III | IV |
|---|---|---|---|---|
| $lnloggrowth_{it}$ | −0.194 *** | −0.227 *** | −0.241 *** | −0.258 *** |
| | (0.021) | (0.031) | (0.020) | (0.014) |
| $lncapital_{it}$ | 0.173 *** | 0.173 *** | 0.131 *** | 0.136 *** |
| | (0.016) | (0.017) | (0.013) | (0.009) |
| $lninflation_{it}$ | −0.230 *** | −0.324 *** | −0.316 *** | −0.255 *** |
| | (0.021) | (0.029) | (0.030) | (0.011) |
| $lnhealth_{it}$ | −0.137 *** | −0.141 *** | −0.158 *** | −0.162 *** |
| | (0.023) | (0.023) | (0.025) | (0.022) |
| $lnChinaFDI_{it}$ | −0.010 | 0.002 | 0.210 | −0.043 *** |
| | (0.142) | (0.166) | (0.191) | (0.005) |
| $lntradeChina\_Africa_{it}$ | 0.018 | −0.014 | −0.023 | −0.087 ** |
| | (0.032) | (0.017) | (0.054) | (0.044) |
| $lninstitution_{it}$ | 0.109 *** | 0.112 *** | 0.190 *** | 0.260 *** |
| | (0.024) | (0.026) | (0.035) | (0.027) |
| $lninstitution*FDI_{it}$ | | 0.016 *** | | |
| | | (0.003) | | |
| $lninstitution*trade_{it}$ | | | 0.034 *** | |
| | | | (0.007) | |
| $lntrade*FDI_{it}$ | | | | 0.035 *** |
| | | | | (0.004) |
| _cons | −3.910 *** | −3.588 *** | −2.613 *** | −3.191 *** |
| | (0.345) | (0.348) | (0.403) | (0.237) |
| Obs. | 448 | 439 | 439 | 439 |
| Groups | 37 | 37 | 37 | 37 |
| Wald (p-value) | 0.000 | 0.000 | 0.000 | 0.000 |
| AB2 (Ch2-sta) p-value | 0.478 | 0.573 | 0.45 | 0.332 |
| Hansen (Ch2sta) p-value | 0.952 | 0.906 | 0.895 | 0.932 |

Notes: *** significant at 1%, ** significant at 5%, standard error in parenthesis, AB2test is Arellano and Bond tests for autocorrelation, Hansen test is a test for over-identification, Wald: overall fitness of the model test, Obs.: observations, Group: group of countries. Per capita real GDP growth of African countries is the dependent variable, $laggrowth_{it}$ represents the lag of growth, $ChinaFDI_{it}$ represents Chinese foreign direct investment in African countries, $tradeChina\_Africa_{it}$ denotes trade between China and African countries, $Capital_{it}$ denotes the physical capital proxied by gross fixed capital formation, $inflation_{it}$ shows the inflation rate of African countries to proxy the macroeconomic stability of African countries, $health_{it}$ indicates the human capital of African countries proxied by gross secondary school enrolment, $institutions_{it}$ is the institutional quality of African countries that proxy domestic the absorptive capacity of countries.

We find almost the same results. The signs of coefficients and their level of significance are almost the same except minor differences in the magnitude and significance of the coefficients of some control variables. The conditional effects of Chinese FDI and the China–Africa trade on the institutional quality of African countries are positive and statistically significant. Likewise, the interaction terms of all China–Africa trade and Chinese FDI is robustly positive. That is the interaction of China–Africa trade openness and FDI to positive and statistically significant, revealing that trade openness between China and African countries fuels the growth effect of Chinese FDI to African countries.

## 6. Discussions

The relationship between foreign trade and economic growth has been theoretically and empirically controversial. While conventional insight predicts a growth-enhancing effect of trade, recent developments suggest that foreign trade is not always beneficial for economic growth.

The empirical analysis in this aspect yields us a mixed conclusion. Some empirical evidence with the positive effect of trade on economic growth (Chemingui and Bchir 2010; Baliamoune-Lutz 2011; Leichenko 2000; Söderbom and Teal 2003; Alcala and Ciccone 2004; Sun and Heshmati 2012) whereas some others with a negative effect and an insignificant effect of trade on economic growth (Ademola et al. 2009; Yin and Vaschetto 2011; Busse et al. 2016). For those who find a positive effect of trade on economic growth, it is essential to emphasize trade relations. For other groups in the literature, the existence of trade relations among countries may not help proper trading nations. However, its positive effect depends on the capability of trading countries to enhance the growth and spillover effect of trade (Chang et al. 2009; Baliamoune-Lutz and Ndikumana 2007; Matthew and Folasade 2014; Zahonogo 2016), indicating that the effects of foreign trade on economic growth are conditional to different the institutional characteristics of the economies.

The results and findings of this study indicate that the China–Africa trade relationship has a robust positive effect on the economic growth of African countries subject to institutional quality. This result on the impact of the China–Africa trade on the economic growth of African countries is inconsistent with some earlier literature, such as Yin and Vaschetto (2011) and He (2013) that have provided evidence that China's increasing trade with Africa is helpful to African sustainable economic growth and development. However, this result is inconsistent with the findings of Ademola et al. (2009), Chemingui and Bchir (2010) and Elu and Price (2010) that reveal for many African countries, the adverse effects of China trade may be greater than the positive ones. Thus, many countries that suffer from insufficient productive capacity and limited economic diversification will not significantly improve, suggesting that trade with China is not a long-term or viable source of development for many African countries.

Despite the theoretical justification that FDI can have a positive impact on the economic growth of host economies through increasing capital accumulation (Solow 1957; Titarenko 2005; Mahembe and Odhiambo 2014), a number of empirical studies found controversial results regarding the effect of FDI on economic growth. More specifically, the empirical studies on the impacts of Chinese FDI on the economic growth of African countries found an unresolved empirical puzzle some with a positive effect (Whalley and Weisbrod 2012; Pigato and Tang 2015; Doku et al. 2017) and others with adverse effects (Foster et al. 2008; Adisu et al. 2010; Busse et al. 2016). Thus, The practical significance of this study, regarding the Chinese FDI–growth nexus, comes at the crossroad of these findings, because examining the effects of FDI on economic growth without incorporating the domestic absorptive capacity of FDI-receiving countries does not provide the full conclusion of the nexus of FDI and economic growth, which means that institutional quality complements Chinese FDI in enhancing the economic growth of African countries. Indeed, the literature has shown that the positive effects of FDI on economic growth are conditioned by the quality of institutions (Ali et al. 2010; Khordagui and Saleh 2013; Jude and Levieuge 2016).

The results of this study further indicate that the growth-effect of Chinese FDI to African countries is negative and statistically significant. However, this finding does not support some previous studies by Doku et al. (2017) and Whalley and Weisbrod (2012), who examined the effect of Chinese FDI on economic growth in Africa and found that the increase in China's FDI into African countries enhances economic growth for African countries. Similarly, the findings of this study do not support the findings of Busse et al. (2016), who find that neither total net FDI inflows nor inflows from China alone have a significant impact on African growth.

This finding on the nexus of Chinese FDI and the economic growth of African countries is not surprising because a number of studies on the FDI–growth nexus came with inconclusive findings. For example, a study by Bruno and Campos (2018) shows that 11% of empirical studies show that FDI has a negative effect on growth, 39% find growth to be independent of FDI and 50% of empirical studies

report a significantly positive effect of FDI on growth. Thus, it seems that FDI plays an ambiguous role in generating economic growth, with little support for an independent, positive effect.

One of the reasons for the controversial outcomes of previous studies might be that previous studies investigated the direct effect of Chinese FDI on economic growth without considering institutional and policy environments of Chinese FDI-hosting African countries. Here, in this analysis, we investigated not only the direct but also the conditional impact of Chinese FDI on the economic growth of African countries that will help observe the full impact of FDI flow.

As a result, as Table 1 indicates, the interaction term of Chinese FDI and institutional quality has a strongly positive effect on the economic growth of African countries. The empirics of this result critically refer that the institutional quality level of a country does help in enhancing the growth effect of Chinese FDI flow to African countries. The positive impacts of Chinese FDI on African countries' economic growth depend on institutional factors such as political stability, the absence of violence, rule of law, regulatory quality and the effectiveness of the government (Asheghian 2004; Hansen and Rand 2006). Furthermore, the finding of this study is consistent with the findings of Adams and OseiOpoku (2015) that the effects of FDI on economic growth is a positive and significant subject to the level of regulations in the FDI host countries. This reveals that the growth effect of FDI is stimulated in the presence of efficient and quality institutions.

These results highlight important potential channels through which FDI may affect growth and are broadly consistent with previous findings on absorptive capacity requirements. Therefore, the findings of this study highlight the complementarity between Chinese FDI to African countries and institutional quality where the impact of FDI on economic growth actually depends on the quality of institutions in the African countries. The findings sum up that the growth-enhancing effect of Chinese FDI on African countries is conditional on the African countries' institutional environment.

From the theoretical point of view, it is stated that FDI can impact the host economies through capital accumulation (Titarenko 2005). For example, one of the neo-classical economic models, the exogenous growth model of Solow (1956, 1957), indicates that economic growth is a function of the labor force and capital, keeping technology exogenous. According to this model, capital accumulation contributes to the economic growth of countries and FDI can contribute to economic growth directly through capital accumulation and expand economic growth (Mahembe and Odhiambo 2014). This study, in this regard, extended this theory to include the mediating role of the domestic institutional quality of African countries in the Chinese FDI and China–Africa trade and economic growth of African countries' relationship. Furthermore, to clearly examine the impact of the China–Africa trade and Chinese FDI, it is better to understand the nature of the economic relationship instruments of China–Africa economic relations. This implies that focusing on the individual pillar of the China–Africa economic relationship cannot explain the full impact of the relationship. Additionally, a policy analysis of the economic relationship between African countries and China should also consider the local institutional environment of African countries.

## 7. Conclusions

This study has examined the impacts of the China–Africa trade and FDI on the economic growth of African countries extending the economic growth model of Mankiw et al. (1992) to include different determinants of economic growth and control China–Africa economic relation indicators. The two-step system GMM model is employed using data for 44 African countries for the periods 2003–2017. The novel part of this study, in this regard, is considering the conditional effect of China–Africa trade and FDI on the economic growth of African countries extending economic relationship theories to include the mediating role of the institutional environment of African countries.

The results of the study reveal that the interaction of Chinese FDI with the institutional quality variable has a significant positive effect on the GDP per capita growth of African countries, indicating the conditional effect of the China–Africa economic relationship on the economic growth of African

countries. Likewise, the conditional effect of China–Africa trade of domestic absorptive capacity on the economic growth of African countries is positive and statistically significant.

Our main conclusion is that institutional quality complements the impacts of China–Africa trade and Chinese FDI on economic growth in African countries. Thus, the results show that Chinese FDI and China–Africa trade alone have no significant positive effect on economic growth in African countries, while a better institutional environment encourages a growth-enhancing effect of Chinese FDI and China–Africa trade to African countries. Thus, the host countries' governments have a crucial role in creating the conditions through improving the level of domestic institutions that consent for the leverage of the helpful effects or for the reduction of the adverse effects of Chinese FDI and China–Africa trade on African countries' economic growth. Therefore, to reap the full benefits from the China–Africa trade and Chinese FDI, considerable improvements in the institutional quality of African economies are required.

**Author Contributions:** Conceptualization, investigation, writing the original draft and formal analysis, M.M.; Validation, writing review & editing and methodology, Q.L.; Formal analysis, data curation, methodology and software, D.G.B.; Investigation, supervision, funding acquisition and resources, J.Y.; Project administration, validation, visualization and methodology, X.Z. All authors have read and agreed to the published version of the manuscript.

**Funding:** This research was funded by Humanities and social sciences fund of the Ministry of Education under grant number 20YJC860006.

**Conflicts of Interest:** The authors declare no conflict of interest.

## Appendix A

**Table A1.** Summary statistics of the variables.

| Variable | Obs. | Mean | Std. Dev. | Min | Max |
|---|---|---|---|---|---|
| Health expenditure | 604 | 5.60 | 2.25 | 1.78 | 20.41 |
| GDP per capita growth | 649 | 2.35 | 6.94 | −62.23 | 122.97 |
| Gross capital formation (in Million USD) | 541 | 10,900.00 | 19,300.00 | 0 | 103,000.00 |
| China–Africa trade | 646 | 1.30 | 2.10 | 0 | 29.03 |
| Inflation (consumer price index) | 623 | 104.04 | 37.23 | 30.42 | 349.82 |
| Chinese FDI to Africa | 657 | 372.55 | 797.28 | 0 | 7472.77 |
| Control of corruption | 660 | −0.58 | 0.59 | −1.63 | 1.22 |
| Government effectiveness | 660 | −0.65 | 0.59 | −1.89 | 1.05 |
| Absence of violence and instability | 660 | −0.46 | 0.83 | −2.67 | 1.20 |
| Regulatory quality | 660 | −0.60 | 0.57 | −2.27 | 1.13 |
| Rule of law | 660 | −0.60 | 0.59 | −1.85 | 1.08 |
| Voice and accountability | 660 | −0.53 | 0.68 | −1.98 | 0.99 |

## Appendix B

**Table A2.** Principal component analysis to derive the institutional quality index.

| Variable | Vector 1 | Vector 2 | Vector 3 | Vector 4 | Vector 5 | Vector 6 | KMO |
|---|---|---|---|---|---|---|---|
| Control of corruption | 0.427 | −0.075 | −0.141 | −0.688 | −0.495 | 0.274 | 0.903 |
| Government effectiveness | 0.422 | −0.395 | −0.216 | −0.045 | 0.721 | 0.312 | 0.866 |
| Absence of violence and instability | 0.356 | 0.805 | −0.415 | 0.197 | 0.080 | 0.093 | 0.937 |
| Regulatory quality | 0.417 | −0.328 | 0.001 | 0.692 | −0.456 | 0.175 | 0.902 |
| Rule of law | 0.444 | −0.114 | −0.066 | −0.075 | 0.035 | −0.882 | 0.866 |
| Voice and accountability | 0.377 | 0.266 | 0.870 | −0.035 | 0.140 | 0.099 | 0.933 |
| Total | | | | | | | 0.896 |

| Component | Eigenvalue | Proportion | Cumulative |
|---|---|---|---|
| Comp1 | 4.725 | 0.788 | 0.788 |
| Comp2 | 0.505 | 0.084 | 0.872 |
| Comp3 | 0.383 | 0.064 | 0.936 |
| Comp4 | 0.210 | 0.035 | 0.971 |
| Comp5 | 0.101 | 0.017 | 0.987 |
| Comp6 | 0.076 | 0.013 | 1.000 |

Multivariate analysis often starts with the data involving a substantial number of correlated variables. Principal component analysis (PCA) is a dimension-reduction tool that can be used to reduce a large set of variables to a small set that still contains most of the information of the large set. It is a mathematical operation that (potentially) converts a number of correlated variables into a (lower) amount of uncorrelated variables called main components. The first main component accounts for as much information variation as necessary, and as much of the remaining variation as possible is accounted for by each successor component.

**Appendix C. African Countries Included in the Sample**

Angola, Benin, Burkina Faso, Botswana, Ivory Coast, Cameroon, Congo Republic, Comoros, Cabo Verde, Djibouti, Algeria, Egypt, Ethiopia, Gabon, Ghana, Guinea, Gambia, Guinea-Bissau, Kenya, Liberia, Libya, Morocco, Madagascar, Mali, Mozambique, Mauritania, Mauritius, Malawi, Namibia, Niger, Nigeria, Rwanda, Sudan, Senegal, Sierra Leone, Seychelles, Chad, Togo, Tunisia, Tanzania, Uganda, South Africa, Zambia and Zimbabwe.

**Appendix D. Chinese FDI to Africa**

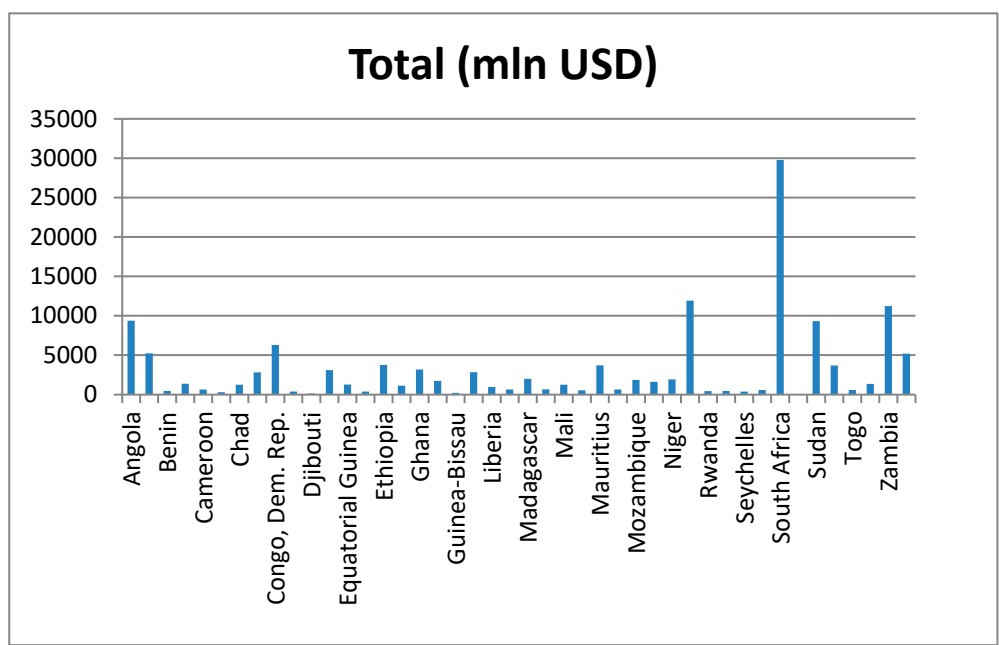

**Figure A1.** Distribution of Chinese FDI in Africa. Source: computed by the authors from UNCTAD Bilateral FDI Statistics, 2017.

## Appendix E. Institutional Quality Performance of Sample African Countries

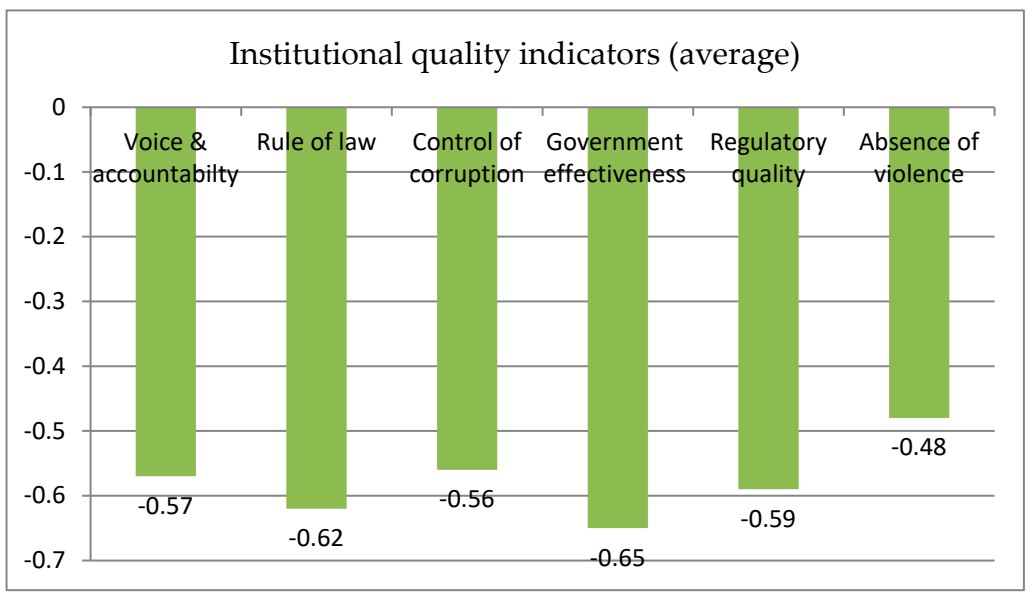

**Figure A2.** Institutional quality performance of sample African countries. Source: computed by authors.

## Appendix F. Average Growth Rates of Sample African Countries

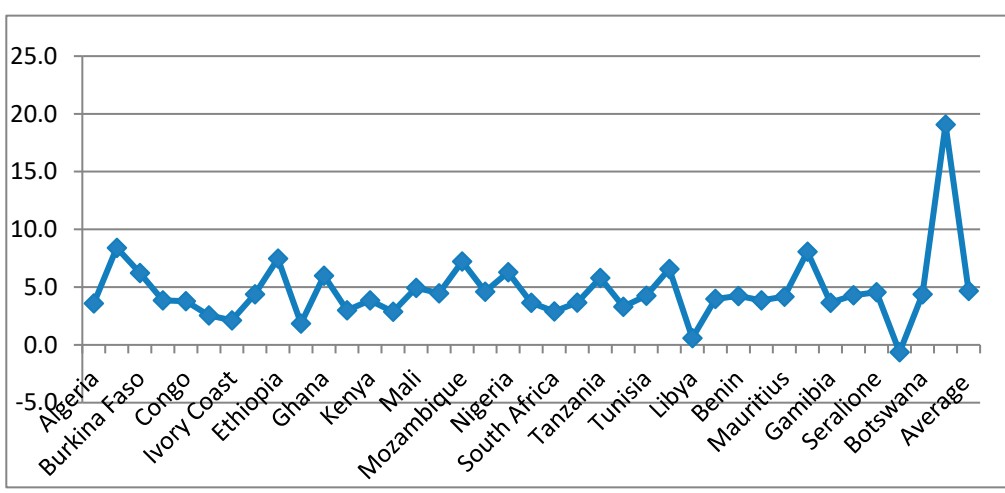

**Figure A3.** Average growth rates of the sample African countries. Source: computed by authors from World Development Indicator Data (WDI).

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
