# Peer review of "The Impacts of Chinese FDI and China–Africa Trade on Economic Growth of African Countries: The Role of Institutional Quality"

_economies, doi:10.3390/economies8030053_

Round 1
Reviewer 1 Report
Very well written paper. I have a few paper suggestions:
- Abstract: please indicate the method used
- Introduction: your introduction should finish with a brief description of the following sections
- line 266: grammatical error instead of "stachastic" it should be "stochastic"
Best of luck!
Author Response
Response to the reviewer 1
We would like to thank the editor for giving us the chance to revise our manuscript. We are also grateful for the reviewer for the detailed evaluation of our paper. The quality of the manuscript has improved after we revised it based on the referee’s insightful suggestions. Our responses are indicated in italics. Revisions in the manuscript are indicated by the red color.
- Very well written paper.
Response: We are grateful for the positive comments.
- Abstract: please indicate the method used.
Response: We are thankful for the suggestion. We have indicated the method used in the study and period covered by the study in the abstract as suggested by the referee (see Lines 9-10).
- Introduction: your introduction should finish with a brief description of the following sections.
Response: thank you for noting it. It has been included in the revised manuscript (see Lines 82-86).
- Line 266: grammatical error instead of "stachastic" it should be "stochastic".
Response: Thank you for noting it. It has been corrected in the revised manuscript (Line 372).
Reviewer 2 Report
Notes to the author(s)
- I found the topic very actual and relevant.
- The motivation, the objectives and the contribution of the article are clearly stated;
- The literature references are recent;
- The methodology is appropriate;
- The results are discussed.
Some points to which I call the author(s) attention:
- Some sentences should be revised. Ex line 228 (“period” and not “periods”); line 229 (the sentence makes no sense) etc; The notes below tables 2 and 3 should be presented with identical format; etc
- The contribution of this paper to the literature on the linkage between foreign FDI and trade on economic growth seem to be the consideration of the role of institutional quality of African countries. If this is the case, I feel that it is too limited. The author(s) should make the contribution of this paper more appealing;
- Being the purpose of this article to assess the impact of Chinese FDI on the economic growth of African economies, this variable should not be classified as a control variable in the models (as it is said in lines 262-263);
- The justification for some results reported on table 3 is not totally convincing. In particular the negative effect of trade;
- In order to explore further the relationship between FDI and economic growth, the author(s) considered the interaction between institutional and FDI and trade, but in those cases the impact of either FDI and trade is not significant. Given these results, the author(s) estimate the model excluding some countries and the results did not change. I consider that these
strategy is not the most adequate to deal with this case in which we have a set of countries
with great differences among them. The authors should consider a methodology to account
for cluster-robust standard errors since the countries they consider are naturally grouped in clusters according to their economic indicators or even their geographic location in the
African continent.
Author Response
Response to the reviewer 2
We would like to thank the editor for giving us the opportunity to revise our manuscript. We are also grateful for the reviewer for the positive feedback and detailed evaluation of our paper. The quality of the manuscript has improved after we revised it based on the reviewer’s insightful revision suggestions. Revision in the manuscript has indicated by the red color. Our responses are indicated in italics.
- I found the topic very actual and relevant.
Response: Thank you for the positive feedback.
- The motivation, the objectives and the contribution of the article are clearly stated. Response: We are grateful for the comments.
- The literature references are recent.
Response: Thank you.
- The methodology is appropriate.
Response: We are remaining grateful for the positive feedback.
- The results are discussed.
Response: We are thankful for the comment.
- Some sentences should be revised. Ex line 228 (“period” and not “periods”); line 229 (the sentence makes no sense) etc; the notes below tables 2 and 3 should be presented with identical format; etc.
Response: We are thankful for noting the errors. They have been corrected in the revised manuscript (see Line 335, Line 388 and Lines 438-447 and Lines 528-537).
- The contribution of this paper to the literature on the linkage between foreign FDI and trade on economic growth seem to be the consideration of the role of institutional quality of African countries. If this is the case, I feel that it is too limited. The author(s) should make the contribution of this paper more appealing.
Response: Thank you very much for the suggestions. The introduction and discussion parts have been broadened to include the contribution of the paper (Lines 72-78, 542-552, and 564-577). Furthermore, we have added section two to support the analysis of the study (lines 89-129).
- Being the purpose of this article to assess the impact of Chinese FDI on the economic growth of African economies, this variable should not be classified as a control variable in the models (as it is said in lines 262-263).
Response: Thank you for noting it. It has been corrected in the revised manuscript (Lines 367-372).
- The justification for some results reported on table 3 is not totally convincing. In particular the negative effect of trade; In order to explore further the relationship between FDI and economic growth, the author(s) considered the interaction between institutional and FDI and trade, but in those cases the impact of both FDI and trade is not significant. Given these results, the author(s) estimate the model excluding some countries and the results did not change. I consider that these strategy is not the most adequate to deal with this case in which we have a set of countries with great differences among them. The authors should consider a methodology to account for cluster-robust standard errors since the countries they consider are naturally grouped in clusters according to their economic indicators or even their geographic location in the African continent.
Response: Thank you for the suggestion. We share your concern that the coefficient of trade is negative. Other econometric specifications which have cluster-robust standard errors are not convenient for the exercise. However, the dynamic model is more convenient in this regard because the analysis of the effect of Africa-China trade and Chinese FDI on African countries' economic growth is based on a pooled data set of cross-country and time-series observations and for reasons discussed in the mythology part we could not run static panel or OLS models with cluster-robust standard errors (see Lines 333-355 for detailed discussion). Furthermore, the GMM estimator demeans the data.
Reviewer 3 Report
The review of the paper titled “The Impacts of Chinese FDI and China-Africa Trade on Economic Growth of African Countries: the Role of Institutional Quality”:
- In the literature review (section 2.1) the authors could more in-depth ground their considerations on novel theories of trade rather than classical (well-known) ones. Generally, in the paper, more attention (and place in the paper) could be put to novel theories instead of signalling classical or neoclassical ones.
- In section 2.2 the authors could also refer to OLI paradigm
- The authors could more comprehensively present the role of foreign ownership in boosting or structural changes in trade, e.g. https://link.springer.com/article/10.1007/s00168-019-00947-6
- The novelty of the research could be written in a more straightforward manner
Author Response
Response to the Reviewer 3
We would like to thank the editor for giving us the opportunity to revise our manuscript. We are also grateful for the reviewer for the positive feedback and detailed evaluation of our paper. The quality of the manuscript has improved after we revised it based on their insightful revision suggestions. Our responses in the manuscript are indicated in red color.
- In the literature review (section 2.1) the authors could more in-depth ground their considerations on novel theories of trade rather than classical (well-known) ones. Generally, in the paper, more attention (and place in the paper) could be put to novel theories instead of signaling classical or neoclassical ones.
Response: Thank you for the comment. This issue has been certainly addressed in the revised version (Lines 162-186, 209-218).
- In section 2.2 the authors could also refer to OLI paradigm. The authors could more comprehensively present the role of foreign ownership in boosting or structural changes in trade.
Response: Thank you for the feedback. It has been discussed in the revised manuscript (Lines 221-229).
- The novelty of the research could be written in a more straightforward manner.
Response: We are thankful for the comment. It has been discussed in detail in the revised manuscript (Lines 72-86, 542-552,564-577).